# Evaluating Self-Concept Measurements in Adolescents: A Systematic Review

**DOI:** 10.3390/children10020399

**Published:** 2023-02-18

**Authors:** Happy Indri Hapsari, Mei-Chih Huang, Maria Wisnu Kanita

**Affiliations:** 1Department of Nursing, College of Medicine, National Cheng Kung University, Tainan 701401, Taiwan; 2Department of Nursing, Kusuma Husada University, Surakarta 57136, Indonesia; 3National Tainan Junior College of Nursing, Tainan 700007, Taiwan

**Keywords:** self-concept, adolescent, psychometric properties

## Abstract

(1) Background: To evaluate the self-concept of adolescents, a proper evaluation of several existing self-concept measurements is needed. The objectives of this study are to conduct a systematic review of the available measures used to assess self-concept in adolescents, to evaluate the psychometric properties of each measurement, and to assess the attributes of patient-reported outcome measurements (PROMs) of self-concept in adolescents. (2) Methods: The systematic review was conducted on six databases: EMBASE, MEDLINE, Cochrane, PubMed, CINAHL, and Web of Science, from inception to 2021. A standardized evaluation of psychometric properties was carried out using the Evaluating the Measurement of Patient-Reported Outcomes (EMPRO). The review was conducted independently by two reviewers. Each attribute in EMPRO was assessed and analyzed to obtain an overall score. Only scores that exceeded 50 were considered acceptable. (3) Results: From 22,388 articles, we reviewed 35 articles with five self-concept measurements. Four measurements were obtained that had values above the threshold (SPPC, SPPA, SDQ-II, and SDQII-S). However, there is not enough evidence to support the interpretability attribute in self-concept measurement. (4) Conclusions: There are various measurements of self-concept in adolescents accompanied by their psychometric properties. Each measurement of adolescent self-concept has a characteristic of psychometric properties and measurement attributes.

## 1. Introduction

Self-concept is a person’s perception of himself/herself that is formed through experience and interpretation of one’s environment. Self-concept is influenced by significant others, praise, or criticism, as well as attributes of one’s behavior [1,2], so the formation of self-concept involves cognitive concepts and memory structures. Self-concept functions as autobiographical memory, an organizer of experience, an emotional barrier, and a source of motivation in a person [3]. In addition, self-concept is multifaceted, which means it involves categorizing large amounts of information [2].

The development of self-concept is influenced by cognitive development. The self-structure that occurs in cognitive development causes individuals to be able to differentiate and integrate. Differentiation allows individuals to be able to carry out self-evaluations in various domains so that they are able to create multiple selves in various contexts. Whereas integration allows individuals to construct higher-order generalizations, namely general self-concepts [4]. Several domains of self-concept in adolescents include the social domain referring to the adolescent’s perception of his role in social relationships, the ability to be socially accepted by others, and skills in appreciating social interactions with others; the physical domain refers to the adolescent’s perception of physical appearance, physical performance, and sports activities; academic domain refers to adolescents’ assessment of academic achievement; emotional domain has the meaning of adolescents’ perceptions of emotional conditions, and responses to specific situations; the domain of morality has components of morality, honesty, and behavior; and general self-concept or self-esteem is an affective action related to the value attached to self-assessment and is shown as a person’s abilities with regard to his strengths and weaknesses [5,6,7]. This affects adolescence, where youths develop and maintain an image as competent, attractive, and valued people through praise and the opinions of their peers [8,9]. Thus, they are integrating various perspectives of the self, for example, how they present themselves to the world, what they want to be, and who they are [3]. Self-concept is a stronger predictors of well-being than educational achievement in adolescents [10].

There were several techniques used for assessing self-concept. Self-report questionnaires or patient-reported outcome measurements (PROMs) were the most commonly used techniques compared to other techniques, such as the semantic differential, an adjective checklist, drawing, or reports from parents or caregivers. A self-report questionnaire can be applied to children, adolescents, and adults who have the ability to read [11]. Of the various kinds of self-concept measurements that have been created, a standard test is needed to select, evaluate, or compare these measurements [12]. Assessment attributes are needed for PROM to achieve a standard in developing PROM that integrates the perspectives of the patient [13,14].

Self-Perception Profile for Children (SPPC), Self-Perception Profile for Adolescents (SPPA), Self-Description Questionnaire II (SDQ-II), Self-Description Questionnaire II short form (SDQII-S), and Piers-Harris Children Self-Concept Scale Second Edition (Piers-Harris 2) are a measure of children’s self-concept which all claim to be valid and reliable measurements [15,16,17,18]. The reliability of the SDQ-II is very impressive, whereas the Cronbach alpha value in several studies is excellent. Likewise, SPPC and SPPA show stability in value over time. In addition, Piers-Harris 2 has carried out language and cultural adaptations with an acceptable process. However, these measurements also have several aspects that are not sufficiently convincing for a psychometric standard of an instrument.

Previously, there was a review of children’s self-concept and self-esteem measurements. However, this review did not use a standardized evaluation tool, and there were no differences between children and adolescents [19]. To date, there has been no systematic review study on measurements of adolescent self-concept.

The Evaluating Measures of Patient-Reported Outcomes (EMPRO) is a standardized and easy-to-use tool for evaluating PROMs. EMPRO was developed based on the latest recommendations of experts, through an expert panel, and refers to the FDA guidelines on the PROM standard. The EMPRO tool was recognized as a reliable, valid tool that can be used in various clinical, administrative, and research settings [14]. EMPRO uses three principal components in psychometric assessment, namely reporting standards, design standards, and statistical outcome standards. This was different from COSMIN which only applies two principal components [12]. In addition, EMPRO has been used in various populations included in systematic review studies, including pathological shoulder disorders [20,21], hemolytic uremic syndrome [22], hip arthroplasty [23], heart failure [24], and prostate cancer [25].

Thus, the objectives of this study are (1) to conduct a systematic review of the available measures used to assess self-concept in adolescents, (2) to evaluate the psychometric properties of each measurement, and (3) to assess the attributes of PROMs of self-concept in adolescents.

## 2. Methods

The method used in this systematic review was based on The Preferred Reporting 82 Items for Systematic reviews and Meta-Analyses (PRISMA) statements [21]. This study was not registered in Systematic Review registration.

### 2.1. Eligibility Criteria

Four key elements were used to determine the eligibility criteria: the construct, the population, the type of measurement, and the measurement properties of the desired construct [26]. In this systematic review, self-concept was defined as a construct, adolescents as a population, patient-reported outcomes as a type of measurement, and psychometric, reliability, validity, reproducibility, or statistical bias as the measurement properties. All articles reporting on the development process of the instrument, its psychometric properties, and administrative issues of self-concept PRO measures were included in this study. Moreover, other inclusion criteria were measurements taken on adolescents aged 10 to 19 years, articles in English, and the availability of full text. Meanwhile, articles reporting case studies, experimental studies, commentaries, letters/reviews/editorials, congress abstracts, and domain-specific such as physical self-concept were excluded from this study.

### 2.2. Information Sources

Most of the articles were obtained from six databases: Embase, MEDLINE, Cochrane, PubMed, CINAHL, and Web of Science, from inception to March 2021. In addition, several articles were obtained that corresponded with the authors of articles or instruments, and Google scholar was used as a secondary source of data.

### 2.3. Search

The search strategy included keywords for the construct of interest (“Self-concept”), population (“Adolescent” OR “Youth” OR “Teen” OR “Female Adolescent” OR “Male Adolescent”), the type of instrument (“Questionnaire” OR “Patient-reported outcome” OR “Surveys and questionnaires” OR “Questionnaires” OR “Patient-reported outcomes”), and the measurement properties of interest (“Psychometry” OR “Reliability” OR “Validity” OR “Reproducibility” OR “Statistical Bias”). The search strategy for each database used in this study is shown in Appendix A (Table A1).

### 2.4. Study Selection

Screening of articles for both title and abstract screening as well as the full-text screening was carried out by HI and MW. If there was disagreement on one or more articles related to the appraisal attributes, an in-depth assessment was carried out. A third reviewer, MC, was consulted if no consensus is reached between the two reviewers.

### 2.5. Data Collection Process

EMPRO is used to measure the performance of patient-reported outcomes measurement to inform the most appropriate candidate among measurements with the same concept [14]. EMPRO consists of 39 items. There were eight attributes: a conceptual and measurement model (concepts and population intended to assess), reliability (to which degree an instrument is free of random error), validity (to which degree an instrument measures what it intends), responsiveness (ability to detect change over time), interpretability (assignment of meanings to instruments’ scores), burden (time, effort and other demands for administration and response), administrative mode of administration (i.e., self- or interviewer-administered, telephone or computer-assisted interview), and cultural and language adaptation (equivalence across translated versions) [14,25,27]. However, the administrative mode of administration attribute was not evaluated because this attribute focuses on how the measurement was filled which was different from the way it was originally designed. Self-concept measurements for adolescents were not found in any other way than self-administered which were filled in by the patients themselves [14].

Each dimension and item on EMPRO were equipped with an explanation to help understand that the appraisal process runs according to the EMPRO standard. The scoring of each item consists of a 4-point Likert scale, with a score of 4 “strongly agree”, and 1 “strongly disagree”. Each item was also given a “no information available” option if the reviewer(s) believed that the supporting documents were insufficient for the reviewer(s) to perform an appraisal on the item. There was also a “not applicable” option for several items in the reliability, validity, and burden domains. This option can be selected if the item cannot be applied to the instrument being evaluated [25,27].

After scoring each item, the reviewer performed coding in the statistics. In the reliability domain, there were two components, namely internal consistency and reproducibility. The highest value of this component was used as a representative of the reliability domain. The value obtained from each item will be calculated on average in each domain, then converted into a score ranging from 0–100. The overall score was the mean of the five dimensions, “conceptual model and measurement”, “reliability”, “validity”, “responsiveness”, and “interpretability” [14,21]. The overall score was calculated if three of the five dimensions have a score. The EMPRO score was accepted if the overall score was at least 50 [14,24].

The data synthesized in this study can consist of qualitative and quantitative data. These data became information in filling out the EMPRO tool, as well as being considered in scoring. Qualitative data can come from the assessment of a panel of experts on self-concept measurement. Focus group discussions can also be used as information to include in the appraisal, especially if it is conducted concerning the target population for aspects of instrument development, content validity, and cultural adaptation of the measurement. On the other hand, quantitative data can include the data distribution, the central tendency score dispersion, ceiling, and floor effects, the pattern of missing data, a factor analysis, Cronbach’s coefficient alpha, a standard error measurement, test-retest reliability, item parameter estimates, criterion validity, and an evaluation of construct validity. In addition, data were also needed through a manual book for using questionnaires, to determine the level of ability of the target population in filling out questionnaires, as well as the skills possessed by administrators.

The EMPRO tool showed highly satisfactory reliability, with a median alpha of 0.95 [14]. Moreover, interrater reliability in this study showed satisfactory results, where the alpha coefficient was 0.966–0.989, and the intraclass correlation value was 0.935–0.978.

### 2.6. Data Items, Risk of Bias, and Synthesis of Results

Data items in each instrument were obtained. If no supporting studies are found for each item, then “no information available” is recorded. The risk of bias in this study used an assessment based on quality, using the criteria defined in the EMPRO. The result was synthesized based on the reported EMPRO attributes in a narrative synthesis.

## 3. Results

### 3.1. Study Selection

Figure 1 shows the flowchart of the review process. We obtained 22,388 articles from six published literature databases. After checking for duplication, a total of 8978 articles were removed. Of the 13,410 articles reviewed for the title and abstract, there were 82 articles for the full-text review. There were several articles excluded because they were not in English (35 articles), were general reviews (9 articles), and did not include the adolescent age category (18 articles). Finally, 36 articles with 5 self-concept measurements in adolescents were included: SPPC, SPPA, SDQ-II, SDQII-S, and Piers-Harris 2. All these questionnaires measure self-concept as a multidimensional construct (Table 1). A measurement, The Five-Factor Self-Concept Questionnaire (AF5), was not included in this study because the articles related to the development process and psychometric properties were in Spanish, and the English version was not available.

### 3.2. Study Characteristics

The findings of the evaluation of the self-concept measurements in adolescents are displayed in Table 1. SPPC was the most published instrument for psychometric properties (33%) when compared to the four other instruments, followed by the SDQ-II (24%) and the SPPA (21%). In addition, the SPPC also had the most published articles after 2010. All these articles were published between 1978 and 2018.

The subscales in the self-concept instruments used on adolescents are shown in Table 2. The subscales were grouped into several categories: academic, social relations, physical, behavioral, other, and overall. A physical appearance subscale was included in these five self-concept instruments.

### 3.3. Findings in Self-Concept Measurements in Adolescents

Table 3 shows the score for each attribute as well as the overall score for each instrument. The conceptual and measurement model attribute refers to the rationale and description of the concept and population to be assessed. In this study, SDQII-S and Piers-Harris 2 have the highest scores, namely 76.19. While SPPC and SDQ-II are in last place with a score of 66.67. The concepts underlying these five measurements are very clearly described. However, not all measures involve the target population in compiling the final content. Only SDQII-S and Piers-Harris 2 have evidence of engaging the target population. SDQII-S involves adolescents with a predominant age of 12–18 years, while Piers-Harris 2 involves adolescents with various ethnicities and socio-economic statuses. Items of measurement scale are also not supported by much evidence. SPPC, SDQ-II, and SDQII-S have relatively low scores when compared to SPPA and Piers-Harris 2.

The reliability attribute is the degree to which the measurement is free from random errors. SDQ-II has the highest rating for this attribute, followed by SPPC and Piers-Harris 2, SPPA, and SDQII-S. SDQ-II, SPPC, Piers-Harris 2, and SPPA reliability scores were obtained from internal consistency scores. There is ample evidence to suggest that data collection methods are internally consistent across these measures. The measurement standard for internal consistency, namely Cronbach’s alpha, shows good results. In SDQII-S, the reliability score is taken from the reproducibility criteria. Data collection methods, the use of retest reliability test designs, as well as interclass correlation coefficients demonstrated satisfactory results.

Validity means the extent to which a measuring instrument measures what it purports to measure, which consists of three types of content-related, construct-related, and criterion-related validity. SPPC, SPPA, and SDQ-II have the highest scores. While SDQII-S has the lowest score, at the minimum threshold. On all five measures, the score related to content validity is low. However, there are quite high scores on construct validity and criterion validity. Although for most of the measurements there is no evidence to suggest that there is a gold standard on criterion validity. All these measurements have been studied in adolescents from various countries, and various ethnicities.

Responsiveness is the ability of a measurement to detect changes over time. SPPC, SPPA, and SDQII-S have high scores, while SDQ-II and Piers-Harris 2 scores are below the threshold score. The criteria for estimating the change in SDQ-II and Piers-Harris 2 are not supported by much evidence, resulting in a low score.

The interpretability attribute refers to the degree to which one can determine an easily understood meaning on a quantitative measurement score. In this study, there are no measurements that exceed the threshold score. Even SPPA and SDQII-S do not have a score because there is no supporting evidence.

Burden refers to the time, effort, and other demands given to those who fill in the measurements (respondent burden), as well as to those who manage the measurements (administrative burden). SPPC and SPPA rank highest in the respondent burden and administrative burden criteria, followed by SDQ-II, and Piers-Harris 2. SDQII-S scores below the threshold on the respondent burden criteria. However, it cannot be assessed on the administrative burden criteria because there is no evidence to support this criterion. These self-concept measurements have been tested on the target population, namely adolescents, and are accompanied by how long it takes to complete them, as well as reading and comprehension levels. Meanwhile, on the administrative burden criteria, there is not much evidence showing special requirements for interviewers in administering these measurements.

The alternative modes of administration attribute are another way of filling out the questionnaire which is different from the design of filling out the original questionnaire. Because there is no other way to fill in the measurements other than self-administered by the patient, this attribute is filled with the “not applicable” option.

Cultural and language adaptation attributes refer to conceptual and linguistic equivalence assessments, as well as the evaluation of measurement properties. Piers-Harris 2 ranks highest, followed by SDQ-II, SPPC, and SPPA. SPPA has a score below the threshold. While SDQII-S cannot be assessed because there is no evidence to support that SDQII-S has been adapted to a culture and language other than English.

Based on the overall score, there are four self-concept measurements in adolescents that exceed the threshold of 50. SPPC has the highest score. The difference between the values for SPPA and SDQ-II is only 0.39 points. The SDQII-S score is 0.24 points higher than the cut-off criterion. Only Piers-Harris 2 falls 0.76 short of the threshold.

## 4. Discussion

Understanding self-concept in adolescents is very important for the formation of a quality young generation [3]. However, appropriate measurements of self-concept are required. In this study, five instruments used for assessing self-concept in adolescents were evaluated, for both instrument development and psychometric scores. These five instruments included, as stated above, the SPPC, SPPA, SDQ-II, SDQII-S, and the Piers-Harris 2. This study is an improvement from previous studies on self-concept review and self-esteem in adolescents [19]. To the authors’ knowledge, there have been no studies investigating self-concept instruments in adolescents.

Four self-concept measurements in adolescents that exceed the threshold score based on the evaluation using EMPRO are SPPC, SPPA, SDQ-II, and SDQII-S. These instruments are easy to use, require a relatively short amount of time, and can be read aloud for teenagers who have difficulty reading due to language differences [15,52].

The conceptual and measurement model attributes showed a score that was not very high. Although the concepts and dimensions of each instrument are explained in detail, the involvement of the target population in the final content is not fully explained, except for the SDQII-S instrument [33]. This may have been because most of these instruments were developed before 1995, except for the SDQII-S, which was developed in 2005. Thus, the standards for instrument development may not have been clear, or the instrument development process may not have been published.

The reliability attribute had a high score on all five instruments. SPPA, SDQ, and PH2 are acceptable in internal consistency. However, several studies showed a fairly low internal consistency value for the SPPA in the romantic appeal domain. This is because some respondents in early adolescence are not yet interested in the opposite sex and romance [41,42,44,45]. Meanwhile, the score on the validity attribute was high. SPPC and SDQ-II were not supported by articles that examined content validity. In addition, no studies used external criteria or estimated the ability of instrument scores to predict relevant concerns, for example, the need for intervention by a healthcare provider.

The manual for each instrument explains in detail the target population and the requirements for administering the instrument. In addition, the manuals also explain how the instrument is given to non-English-speaking teenagers. However, the completion rate of the Piers-Harris 2 was lower than that of the SPPC, at 50% and 79%, respectively. With a dichotomous response item (Yes/No), the respondent can give another answer, such as “sometimes” or “middle”. This indicates that the answer choices are too limited [33].

SPPC was tested psychometrically on various populations, including students [15,28,29,30,31,33,39], academically talented students [35], psychiatric patients [40], adolescents with various ethnicity [36,37], and adolescents with chronic diseases [32]. This is different from the psychometric properties of the SDQ-II, SDQII-S, SPPA, and Piers-Harris 2, which are only used to test healthy students [16,17,18,28,41,43,44,45,47,48,49,50,51,55,58], gifted students [35,54], adolescents with various ethnicity [42], and indigenous secondary school students [56,57].

Besides having been tested on various types of populations, several measures of self-concept have also been tested based on gender. This is because gender differences in adolescence contributed to the domains of self-concept [60]. SDQ-II was an instrument that has been tested for psychometric properties in adolescent boys and girls with almost the same proportion [47,48,49,50,51,54,55]. SPPC has also been tested on both genders [15,28,29,30,31,32,33,35,38,39,40], although there were studies that focused only on African American girls [36,37]. Moreover, Piers-Harris 2 tested the psychometric properties of both genders [18,58]. Different things were found in the SPPA and SDQII-S instruments which report some psychometric properties tests without including gender in the research sample [17,42,44,56].

The SPPC and SDQ-II have been adapted for language and show acceptable psychometric values. SPPC has been translated into Dutch [39,40], Arabic [29], and Portuguese [31]. The SDQ-II has been translated into Swedish and Japanese [55], Spanish [50], and French [49]. Nevertheless, the authors did not use the latest guidelines on translation, adaptation, and validation of instruments [61,62].

The SPPA has been adapted into several languages and cultures, such as Norwegian and Australian. The job competence domain was omitted from the instrument because it was not the same in the two countries [43,44]. Studies conducted in America also show a great deal of missing data in the job competence and romantic appeal domains when compared to other domains [45]. Moreover, the romantic appeal domain has been changed to “American Dating” after cultural adaptation [44].

Our study has several limitations. First, we limited the instrument to only instruments developed using English, which excluded some non-English articles such as The Five-Factor Self-Concept Questionnaire (AF5), which was developed in Spanish. Second, although we used multiple databases (Embase, MEDLINE, Cochrane, PubMed, CINAHL, Web of Science), we did not search the psychology database, PsyINFO, due to limited access to that database. Self-concept is closely related to psychology, and this is one of our concerns from the very beginning when we searched for articles. Third, we screened the age of the population from the screening title/abstract, so it was not possible for us to extract data purely from adolescents based on these articles.

## 5. Conclusions

This study is the first to evaluate self-concept instruments in adolescents. Available evidence supports that the SPPC, SPPA, SDQ-II, and SDQII-S are self-concept instruments for adolescents with scores over the threshold, based on EMPRO evaluation, although there is still no complete information covering all aspects of using these instruments. Based on the assessed attributes, we emphasize that in developing an instrument, especially a PROM, holding panel discussions with experts and the target population is very important. This also supports the validity of the content of the instrument of interest. In addition, researchers should refer to cultural guidelines and language adaptation for translating, adapting, and validating self-concept instruments in children/adolescents. It is necessary to begin developing computer-based instruments to ease the participant burden, as well as accelerate data collection and analysis.

Although the score is below the threshold, the Piers-Harris 2 is a good instrument based on the attributes of cultural and language adaptation. Psychometric properties should also be tested in various cultures and languages, as well as various types of populations, not just focusing on healthy adolescents. Hence, a more comprehensive psychometric study can be obtained.

## Figures and Tables

**Figure 1 children-10-00399-f001:**
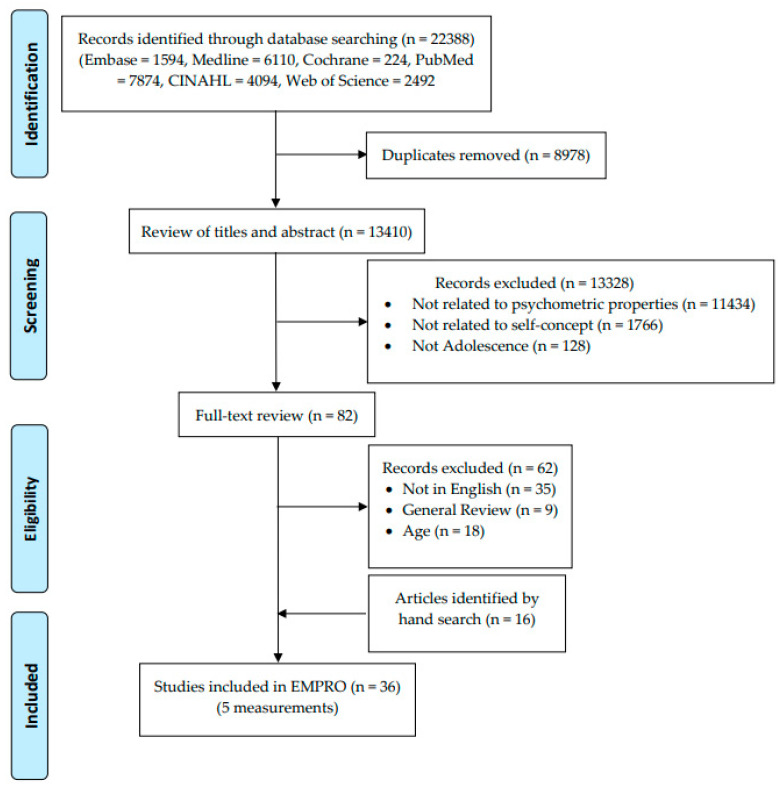
Flowchart of review process.

**Table 1 children-10-00399-t001:** Summarized characteristics of the instruments identifying self-concept in adolescents.

Instrument	Author (Year)	No. of Items	Scales	Ages for Population	Time to Complete	No. of Articles
SPPC	Harter, S. (2012)	36	4-point Likert scale. Scored 6–24 for each domain.	Aged 8–15; Grades 3 to 7 or 8	-	14 articles (33%) [15,28,29,30,31,32,33,34,35,36,37,38,39,40]
SPPA	Harter, S. (2017)	45	4-point Likert scale. Scored 5–20 for each domain.	Aged 13–18; Grades 8 to 12	50 min	9 articles (21%) [16,28,34,35,41,42,43,44,45]
SDQ-II	Marsh, H.W. (1992)	102	4-point Likert scale. Raw score converted to 0–100	Aged 13–17; Grades 7 to 10	20–25 min	10 articles (24%) [46,47,48,49,50,51,52,53,54,55]
SDQII-S	Marsh, H.W. et al. (2005)	51	4-point Likert scale. Raw score converted to 0–100	Aged 13–17; Grades 7 to 10	-	6 articles (14%) [17,46,52,53,56,57]
Piers—Harris 2	Piers and Herzberg (2002)	60	Dichotomy.	Aged 7–18	10–15 min	3 articles (7%) [18,58,59]

**Table 2 children-10-00399-t002:** Subscales of the self-concept instruments used for adolescents.

Subscale		SPPC	SPPA	SDQ-II	SDQII-S	Piers-Harris 2
Academic	Scholastic Competence	X	X			
	Math			X	X	
	Verbal			X	X	
	General School			X	X	
	Intellectual and School Status					X
Social Relationship	Social competence	X	X			
	Job competence		X			
	Romantic appeal		X			
	Close friendship		X			
	Parent relationship			X	X	
	Same-sex relationship			X	X	
	Opposite-sex relationship			X	X	
	Popularity					X
Physical	Athletic competence	X	X			
	Physical appearance	X	X	X	X	X
	Physical ability			X	X	
Behavior	Behavioral adjustment					X
	Behavioral conduct	X	X			
Psychological	Emotional stability			X	X	
	Honesty and trustworthiness			X	X	
	Anxiety					X
	Happiness					X
Overall	Global self-worth	X	X			
	General			X	X	

X: the domain contained in the questionnaire; Instruments: SPPC is the Self-Perception Profile for Children; SPPA is the Self-Perception Profile for Adolescents; SDQ-II is the Self-Description Questionnaire II; SDQII-S is the Self-Description Questionnaire II short form; Piers-Harris 2 is the Piers-Harris Children Self-Concept Scale Second Edition.

**Table 3 children-10-00399-t003:** Ratings of each EMPRO item and attributes for the instruments used to evaluate adolescents’ self-concept.

Attributes	SPPC	SPPA	SDQ-II	SDQII-S	Piers-Harris 2
Conceptual and Measurement Model	66.67	71.43	66.67	76.19	76.19
1. The concept is clearly stated	++++	++++	++++	++++	++++
2. Obtaining and combining items into dimensions is described	++++	++++	++++	++++	++++
3. The dimensionality is described	++++	++++	++++	++++	++++
4. Obtaining final content involve target population	++	++	++	++++	+++
5. Scale variability in the population	++	++	++	++	++
6. The measurement data scale is clearly explained	++	+++	++	++	+++
7. The total score of the raw score is clearly explained	+++	+++	+++	+++	+++
Reliability	83.33	66.67	91.67	58.33	83.33
Internal Consistency	83.33	66.67	91.67	50	83.33
8. Data collection methods for internal consistency are explained	+++	+++	+++	+++	+++
9. Cronbach alpha coefficient value, KR-20	+++	++++	++++	+++	++++
10. IRT approach	++++	NA	++++	NA	+++
11. Populations that can affect internal consistency	++++	++++	++++	+++	++++
Reproducibility	NA	NA	33.3	58.33	NA
12. Data collection methods for reproducibility are explained	NA	NA	+++	++++	NA
13. Clear reasons for the test-retest design and time interval	NA	NA	++	+++	NA
14. Value of the interclass correlation coefficient (ICC)	NA	NA	++	+++	NA
15. Item parameter estimates	NA	NA	+	NA	NA
Validity	66.67	66.67	66.67	50	53.33
16. Evidence for content-related validity	+	++	+	+	+
17. Methods for construct- and criterion-related validity	+++	+++	+++	+++	+++
18. Sample for construct- and criterion-related validity	++++	++++	++++	+++	+++
19. Availability of hypotheses in the construct validity	+++	++	++++	NA	++
20. Gold standard in criterion validity	NA	NA	NA	+++	NA
21. Population in testing validity	++++	++++	+++	++++	++++
Responsiveness	100	100	44.4	66.67	33.3
22. Methods for testing responsiveness	++++	++++	++++	++++	+++
23. Estimation of the magnitude of change	++++	++++	+	++	+
24. The group used in testing the magnitude of change.	++++	++++	++	+++	++
Interpretability	0	NA	22.2	NA	0
25. Reasons for determining external criteria	+	NA	+++	NA	+
26. Interpretation strategy	+	NA	+	NA	+
27. Interpretability data presentation	+	NA	+	NA	+
Burden					
Respondent Burden	77.78	77.78	66.67	44.4	55.56
28. Requirements of respondents to fill out the questionnaire	+++	+++	++++	++	+++
29. Acceptability of instrument	+++	+++	+	+	+
30. Circumstances about the acceptability of the instrument	++++	++++	++++	++++	++++
Administrative Burden	88.89	88.89	75	NA	61.11
31. Resources in administering instruments	++++	++++	++++	NA	+++
32. Time required to administer the instrument	NA	NA	NA	NA	NA
33. Instrument administrator requirements	NA	NA	NA	NA	NA
34. Information regarding instrument scoring	++++	++++	+++	NA	+++
Alternative Modes of Administration	NA	NA	NA	NA	NA
35. Characteristics of each alternative mode of administration	NA	NA	NA	NA	NA
36. Comparison of each alternative mode of administration	NA	NA	NA	NA	NA
Cultural and Language Adaptation	66.67	55.56	77.78	NA	88.89
37. Linguistic equivalence test	++	++	+++	NA	++++
38. Methods of conceptual equivalence	+++	+++	+++	NA	+++
39. The difference between the original version and the adapted version	++++	+++	++++	NA	++++
Overall Score					
	63.33	60.95	60.56	50.24	49.24

Description: + 1 (strongly disagree); ++ 2; +++ 3; ++++ 4 (strongly agree); NA not applicable. Instruments: SPPC: the Self-Perception Profile for Children; SPPA: the Self-Perception Profile for Adolescents; SDQ-II: the Self-Description Questionnaire II; SDQII-S: the Self-Description Questionnaire II short form; Piers-Harris 2: the Piers-Harris Children Self-Concept Scale Second Edition.

## Data Availability

Not applicable.

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
