# Peer review of "Evaluating Self-Concept Measurements in Adolescents: A Systematic Review"

_children, 2023, doi:10.3390/children10020399_

Round 1

Reviewer 1 Report

First of all I would like to thank you for the opportunity to have reviewed this research. I believe that for the publication of the study it is necessary to carry out a greater contextualisation in the theoretical section, as well as to include more current bibliographical references. 

Regarding the objective of the study, I think it is answered in a concise way. 

Table number two would be marked with an X instead of the current symbols, as this would improve the presentation of the study. 

Author Response

Point 1: I believe that for the publication of the study, it is necessary to carry out a greater contextualization in the theoretical section, as well as to include more current bibliographical references.

 Response 1:

Thank you for your critical comments. We added some references to enrich the theory of adolescent self-concept in the introduction section, lines 39-53, and 57-58. The references we use are between 2016 and 2022, although there is one reference in 2006.

In addition, in this manuscript, we still use references that are quite old, because they are related to the development of self-concept measurement. Although SPPC and SPPA were written in 2012 and 2017 respectively, their first creation began in 1982 under the title The Perceived Competence Scale for Children. Likewise, SDQ-II which was written in 1992, with a short version made in 2005.

Point 2: Regarding the objective of the study, I think it is answered in a concise way.

Response 2: Thank you for the kind words.

Point 3: Table number two would be marked with an X instead of the current symbols, as this would improve the presentation of the study.

Response 3: Thank you for your suggestion. We realized our mistake in using graphic images that did little to support our study. We have replaced it with an "X" according to your suggestion (Table 2, line 243).

Reviewer 2 Report

Thank you for the opportunity to review the manuscript entitled, “Evaluating self-concept measurements in adolescents: A systematic review.”

 Below I provide my overall impressions followed by more specific comments.

OVERALL COMMENTS:

1.     This manuscript draws attention to an important topic in self-concept scales in adolescents.

2.     Parts of this manuscript could be strengthened. For example, including more details on rationals to select five measures used for analysis and in the results section are needed.   

Why choose these five measures and exclude others? also mention which scales are not selected and exclusion criteria. 

3.     The authors did a good job identifying relevant research to support this study. Some areas need additional citations.

4.  Please provide more detailed information regarding analyses tool used in this study.

Author Response

Point 1: This manuscript draws attention to an important topic in self-concept scales in adolescents.

 Response 1: Thank you for the positive comment.

Point 2: Parts of this manuscript could be strengthened. For example, including more details on rationales to select five measures used for analysis and in the results section are needed. Why choose these five measures and exclude others? Also mention which scales are not selected and exclusion criteria.

Response 2:

Thank you very much for your comments.

  • We have revised the inclusion and exclusion criteria in the eligibility criteria section, lines 109-115. We added "all articles reporting on the process of developing the instrument, its psychometric properties, and administration issues on the self-concept instrument with the adolescent population". Thus, we have included articles that fully present how the instrument was developed or tested in the context of psychometry in selected populations.
  • We also revised the results section, lines 196-199. We added information about measuring self-concept using AF5 which we excluded due to the unavailability of English language AF5 instruments, along with articles related to psychometry. In addition, we have also written about AF5 measurement in the limitations of our study.

Point 3: The authors did a good job identifying relevant research to support this study. Some areas need additional citations.

Response 3: Thanks for the very detailed advice. We added a reference citation about EMPRO as an appraisal tool in the data collection process section (method section), lines 137-165.

Point 4: Please provide more detailed information regarding analysis tool used in this study.

Response 4: Thanks for your advice. We have added information regarding EMPRO as an appraisal tool in the data collection process section, lines 137-144.

Reviewer 3 Report

Adolescent self-concept is an important construction when attempting to understand and describe behavior among adolescents.  This paper utilizes a well-designed approach to measuring and comparing adolescent self-concept instruments, and includes a comprehensive set of criteria on which these comparisons were made. 

The initial number of articles included in the study was very extensive and the methodology for inclusion/exclusion was quite rigorous.  The paper does a good job of describing the step-by-step approach to determining which articles would remain for the final psychometric comparisons.  Readers should be able to follow the procedural guidelines used for determining the final inclusion of articles and for identifying all of the specific measures used in the final comparisons.

Overall, the paper is well written and addresses an important and interesting topic.  I believe that the journal's audience will enjoy reading this paper.

Author Response

Point 1: Adolescent self-concept is an important construction when attempting to understand and describe behavior among adolescents.  This paper utilizes a well-designed approach to measuring and comparing adolescent self-concept instruments, and includes a comprehensive set of criteria on which these comparisons were made.

 Response 1: Many thanks for your positive words for us.

Point 2: The initial number of articles included in the study was very extensive and the methodology for inclusion/exclusion was quite rigorous.  The paper does a good job of describing the step-by-step approach to determining which articles would remain for the final psychometric comparisons.  Readers should be able to follow the procedural guidelines used for determining the final inclusion of articles and for identifying all of the specific measures used in the final comparisons.

Response 2: Thank you in advance for the positive comments

Point 3: Overall, the paper is well-written and addresses an important and interesting topic.  I believe that the journal's audience will enjoy reading this paper.

Response 3: Thank you for your trust in our study.

Reviewer 4 Report

During the analysis of the materials of this study, presented in the article “Evaluating Self-Concept Measurements in Adolescents: A Systematic Review”, I come to the conclusion that the article title and abstract are appropriate.

The purpose of the article and its significance is stated clearly. The method used in this systematic review was based on The Preferred Reporting Items for Systematic reviews and Meta-Analyses (PRISMA) statements. The writing is clear and concise. The conclusions are accurate and supported by the content. The article is of interest to members of the education research community.

I recommend A Systematic Review “Evaluating Self-Concept Measurements in Adolescents” for publication on the pages of the International Journal “Children” (ISSN 2227-9067).

MD, Professor,

S.Igumnov  /Sergey Igumnov/

Author Response

Point 1: During the analysis of the materials of this study, presented in the article “Evaluating Self-Concept Measurements in Adolescents: A Systematic Review”, I come to the conclusion that the article title and abstract are appropriate.

 Response 1: Thank you for the detailed comments.

Point 2: The purpose of the article and its significance is stated clearly. The method used in this systematic review was based on The Preferred Reporting Items for Systematic reviews and Meta-Analyses (PRISMA) statements. The writing is clear and concise. The conclusions are accurate and supported by the content. The article is of interest to members of the education research community.

Response 2: Thank you for your opinion.

Point 3: I recommend A Systematic Review “Evaluating Self-Concept Measurements in Adolescents” for publication on the pages of the International Journal “Children” (ISSN 2227-9067)

Response 3: Thank you very much for your recommendations for this manuscript. We hope that this study will provide many contributions to future self-concept research on adolescents.

Round 2

Reviewer 1 Report

The article has been improved so it can be publish now

Author Response

Point 1: The article has been improved so it can be publish now

Response 1:

Thank you very much for your kind words.
